# Structural, Optical, Charge-Transport, and Dielectric Properties of Double-Perovskite La_2_Co_1−z_Fe_z_MnO_6_ (z = 0, 0.2–1.0)

**DOI:** 10.3390/ma15186249

**Published:** 2022-09-08

**Authors:** Ghulam Hussain, Shanta Batool, Yuruo Zheng, Shuyi Li, Xiawa Wang

**Affiliations:** 1Department of Natural and Applied Sciences, Duke Kunshan University, Kunshan, Suzhou 215316, China; 2Department of Physics and Key Laboratory of Strongly-Coupled Quantum Matter Physics (CAS), University of Science and Technology of China, Hefei 230026, China

**Keywords:** multiferroics, infrared reflectivity, conduction mechanism, dielectric response

## Abstract

A series of double-perovskite La_2_Co_1−z_Fe_z_MnO_6_ (z = 0, 0.2–1.0) ceramics were synthesized using a well-established sol–gel method. The series of samples with a monoclinic phase and a P2_1_/n symmetry were characterized by XRD, FTIR, conductivity, and capacitance measurement to extract charge-transport and dielectric characteristics at room temperature. The obtained IR spectra fitted well with the Lorentz oscillator model to calculate the damping factor, optical frequency, and oscillator strength and compared with the theory, which gave better agreement. The calculated activation energies from the Arrhenius plot supported the semiconducting nature of all samples. The temperature and frequency-dependent dielectric parameters, such as the real part ($\varepsilon_{r}^{\prime }$), imaginary part ($\varepsilon^{\prime \prime }$) of the dielectric constant, dielectric loss (tanδ), and ac-conductivity (*σ*_ac_) were extracted. The dielectric constant ($\varepsilon_{r}^{\prime },\;\;\varepsilon^{\prime \prime }$) and dielectric loss (tanδ) were enhanced at a low frequency, while the ac-conductivity (σ_ac_) displayed higher values at higher frequencies. The enhancement in the dielectric parameters with increasing iron concentrations arose due to the higher surface volume fraction of iron (Fe^3+^) ions than the cobalt (Co^3+^) ions. The radius of the Fe^3+^ (0.645 Å) was relatively higher than the Co^3+^ ions (0.61 Å), significantly influenced by the grains and grain boundaries, and enhanced the barrier for charge mobility at the grain boundaries that play a vital role in space charge polarization.

## 1. Introduction

Over the last few decades, the field of multiferroics has attracted enormous attention from the scientific community owing to their unique physical properties, which hold great importance in practical applications [1,2,3]. Due to strong electron correlations in these materials, they show such effects as magnetism, ferroelectricity, multiferroic properties, or superconductivity. Several competing properties, such as magnetic order and electric polarization, often coexist [4,5,6]. In memory-storage devices, controlling magnetic components through electric manipulation will provide faster data storage and recovery. The merger of the magnetic order, switchable properties, and electric polarization on one data-storage device will increase the storage size by five orders of magnitude. Thus, the area currently required for 1 megabyte of information will be capable of carrying one terabyte. Understanding the physical effects behind these phenomena is essential to allow a systematic search for new composites with these combined properties.

The well-known ferroelectrics, including KNbO_3_, BaTiO_3_, BiFeO_3_, and PbTiO_3_, belong to first-class multiferroics. A few boracites with chemical formula M_3_B_7_O_13_X, with M and X representing the divalent metal and halogen, have tetrahedral phase ferroelectrics [5,7,8] Owing to the coexistence of ferromagnetism and electrical insulation within a single material, ferromagnetic materials with chemical formula RE_2_AMnO_6_, where RE and A correspond to the rare-earth element and transition metal element, respectively, attract a lot of interest. Of particular note, the double-perovskite La_2_CoMnO_6_ in all forms, i.e., thin films, single crystals, and polycrystalline compounds, has attracted considerable attention because of its intriguing magnetodielectric feature and spintronic device applications, such as electronic memories or charge-storage devices [9]. Ferromagnetic La_2_CoMnO_6_ also exhibits the ordered structure of double perovskite with MnO_6_ and NiO_6_ octahedra aligned in the composition of rock salt. The super-exchange interactions in ordered form Co^2+^-O-Mn^4+^ offer great ferromagnetism with a transition temperature (T_c_) value around 280 K. Recent works focused on the magnetic properties of La_2_CoMnO_6_, with B-site atomic ordering Co/Mn and cationic oxidation states in the form of Co^2+^/Mn^4+^ or Co^3+^/Mn^4+^ [10,11,12,13]. The radius of the Mn^3^^+^ ion is 0.65 Å and Fe^3^^+^ ion 0.645 Å in a high-spin state, which are relatively higher than the Co^3^^+^ ion (0.61 Å) in a low-spin state. However, Mn^3^^+^-O^2^-Fe^3^^+^ is expected to have one of the most robust ferromagnetic exchange interactions, and its atomic arrangement (e.g., disorder) has a remarkable impact on the physical properties of La_2_FeMnO_6_. Recently, the controlled substitution, replacing the atom with almost similar ionic size, and search for new dopants have drawn intensive research attention to improving the double-perovskite La_2_Co_1−z_Fe_z_MnO_6_ structure and physical properties due to its importance both in terms of understanding of fundamental physics and potential for device applications. Qiuhang Li et al. synthesized the double-perovskite La_2_Co_1−x_Fe_x_MnO_6_ and explored the transport properties [14]. Mohd Nasir et al. reported that relative alteration in the Ni:Mn ratio strongly influenced the structural and magnetic properties of La_2_NiMnO_6_ [15]. Moreover, J. Krishna Murthy et al. replaced Ni with Co substitution in La_2_Ni_1−x_Co_x_MnO_6_ (0 < x < 1) nanoparticles’ and explored the magnetic properties. [16]. Hui Gan et al. fabricated Co-doped La_2_NiMnO_6_ (La_2_Co_x_Ni_1−x_MnO_6_, x = 0.1–0.5) ceramics at low temperatures and improved the magnetic performance [17]. However, such efforts have been explored in studies of even broader physical phenomena. In this regard, we synthesized La_2_Co_1−z_Fe_z_MnO_6_ (z = 0, 0.2–1.0) ceramics at room temperature.

This work reports the phase, optical, charge-transport, and dielectric properties of La2Co1−zFezMnO6 (z = 0, 0.2–1.0) ceramics at room temperature. A series concentration of iron doping with the replacement of Co+3 ions is measured. Fe3+ is a kind of magnetic transition metal ion, which plays a vital role in determining the physical and dielectric properties of double-perovskite La2Co1−zFezMnO6 ceramics. This work helps find a systematic search for new composites by exploring the physical effects behind these phenomena.

## 2. Experiment

The series of double-perovskite La_2_Co_1__−__z_Fe_z_MnO_6_ (z = 0, 0.2–1.0) ceramics were synthesized by the well-established sol–gel process as described in [18] with all chemical purities higher than 99%. All samples were sintered for 6 h at 950 °C to obtain the required phase. We used an advanced diffractometer (Bruker, D8 DISCOVER, Billerica, MA, USA with Cu-Kα (λ = 1.5405 Å) radiation source) to verify the structure of ceramics, the phase was identified by Xpert HighScore software, and Rietveld refinement was carried out with the help of JANA 2006 software. Furthermore, using an Fourier-transform infrared (FTIR) spectrometer (Bruker, Vertex 80v, Billerica, MA, USA) close to normal incidence mode, infrared reflectivity (IR) spectra were determined. Diffuse reflectance spectra with a range of 30–7500 cm^−1^ were obtained from the spectrophotometer (PerkinElmer, Lambda 950 UV/VIS/NIR, Waltham, MA, USA) with a two-beam splitter detector. The electric and dielectric properties were tested with an LCR meter (GW Instek, 8101G, Taiwan, China) in a temperature range of 303–423 K and frequency range from 10 Hz–1 MHz. The charge-transport properties were performed using a Keithley meter (Keithley Instruments, 2400, Cleveland, OH, USA) at various temperatures.

## 3. Results and Discussion

### 3.1. Structural Properties

The X-ray diffraction pattern confirmed the structure of the double-perovskite La_2_Co_1−z_Fe_z_MnO_6_ ceramics at room temperature. Further investigation confirmed that this type of compound has a monoclinic phase with a P2_1_/n symmetry at room temperature. With the help of the fitting software, XRD peaks are well indexed, indicating that all series of this compound have a monoclinic phase with a P2_1_/n space group with a slight increase in the lattice parameters. The maximum intensity peak is observed at 2θ ≈ 32°, representing the principal peak of the P2_1_/n space group. All peaks are well fitted to the calculated pattern, indicating that the double-perovskite La_2_Co_1−z_Fe_z_MnO_6_ has a single phase that displays the purity of the ceramics shown in Figure 1.

The Braggs peaks were slightly shifted towards the right with more iron substitution because the ionic radius of the Fe^+3^ (0.645 Å) ion is relatively larger than the Co^+3^ (0.61 Å) ion. The larger ionic radius caused expansion or shrinkage of the structure and a slight increase in the lattice parameters a, b, and c. Due to the variation in particle size, the surface-to-volume ratio also fluctuated. The obtained refined lattice-parameter values are listed in Table 1, which are in agreement with the parameters reported by others [19].

The plot of the lattice parameters versus the contents is shown in Figure 2. We found that the volume and grain size fluctuated with the increase in Fe^+3^ concentrations during refining the structural parameters. The average crystalline grain size of La_2_Co_1−z_Fe_z_MnO_6_ (z = 0, 0.2–1.0) estimated the (2 0 0) peak with the help of the Scherrer formula given by Equation (1), and the values are listed in Table 1 [12].
(1)$$D=\frac{k\lambda }{\beta \cos\theta }$$
where *k* represents the geometry factor, and its value is considered as 0.9, *θ* is a diffracted Braggs angle, and *β* defines the full width at half maximum.

### 3.2. Optical Properties

The IR reflectivity spectra of the series of double-perovskite La_2_Co_1−z_Fe_z_MnO_6_ (z = 0, 0.2–1.0) ceramics were measured using FTIR at room temperature, as shown in Figure 3 and Figure 4. There were 24 Raman-active modes (12 A_g_ + 12 B_g_) and 33 infrared reflectivity (IR) active modes (Г_IR_ = 17 A_u_ + 16 B_u_) for the monoclinic phase according to the group theory [13]. We conducted IR tests, and only 12 active phonon modes were observed in our samples instead of 33 phonons in La_2_Co_1−z_Fe_z_MnO_6_. We did not observe other active phonon modes due to the polycrystalline nature of the material, in which grains have different orientations. However, our samples exhibited the monoclinic phase, but we observed less polar phonons modes than the theoretically predicted 33 polar phonon modes for this structure. For the searching of optical phonons mode, we refined our experimental results with the help of the Lorentz oscillator model. The equation of the dielectric function and IR via Fresnel’s law are expressed in Equations (2) and (3).
(2)$$R\left(\omega \right)={\left|\frac{1+\sqrt{\varepsilon \left(\omega \right)}}{1-\sqrt{\varepsilon \left(\omega \right)}}\right|}^{2}$$
(3)$$\varepsilon \left(\omega \right)=\varepsilon_{\infty }+{\frac{\sum_{j}\omega_{TO\left(j\right)}^{2}s_{j}}{\omega_{TO\left(j\right)}-\omega^{2}-i\omega \gamma }}_{j}$$
where *ϒ**_j_* and *S_j_* are the damping factor and the oscillator strength of the *j*-th optical phonon, *ε*_∞_ displays the charge polarization, and $\omega_{TO\left(j\right)}^{2}$ represents the *j*-th optical transverse phonons [18,20]. The fitted parameters of optical transverse phonon values of La_2_Co_1−z_Fe_z_MnO_6_ (z = 0, 0.2–1.0) are listed in Table 2. The parameters of the phonon oscillator strength amplitude “*S*” are listed in Table 3. The damping factor “*γ*” with the width of the vibrational oscillation phonon mode is inserted in Table 4. In the low-frequency region, the observed IR spectra of La_2_Co_1−z_Fe_z_MnO_6_ display that the phonon modes are slightly shifting with the concentration of Fe^+3^. Two reasons potentially caused the shifting of frequency below 200 cm^−1^ are mass substitution effects and the constant force described by the harmonic oscillator equation.
(4)$$\omega =\sqrt{D/u}$$
where D represents the constant force that depends on the bond length, bond angle, and lattice parameters. This constant force is highly sensitive to the bond length and lattice parameters. u represents the reduced mass of those ions, which contributes to the related phonon modes. Hence, the phonon frequency will be enhanced or suppressed due to the variation of constant lattice parameters. Due to the higher surface volume fraction, Fe^3+^ ions’ influence on the optical transverse mode causes an increase in the phonon frequency.

The intermediate phonon is observed above 200 cm^−1^ due to the motion of B ions against the vibrations of oxygen (MnO_6_). At higher frequencies (>500 cm^−1^), the phonon modes are attributed to the stretching of the octahedral (Fe/Mn)O_6_. In the low-frequency range ≤200 cm^−1^, one phonon ω_TO1_ is noted at 127.10, 124.83, 130.83, 129.10, 128.75, and 127.40 cm^−1^ for (z = 0, 0.2–1.0), which shows the altering of phonon modes due to the reduced mass effect of Fe^+3^. These parameters are confirmed from the harmonic oscillator relation corresponding to the force constant and reduced mass. In the low-frequency region, the phonons (ω_TO4_) modes are missing for the z = 0.8 and 1.0 contents, which may be shifting of phonon modes due to the reduced mass effect of magnetic metal ions (Fe^+3^ and Co^+3^). In the high-frequency region (400–700 cm^−1^), the phonon mode ω_TO9_ is missing for z = 1.0. Due to the stretching of phonon modes or octahedral distortions of MnO_6_ in the higher frequency regions above (400–700 cm^−1^), the exploration of optical phonon modes is interesting. The structural distortion is produced due to the reduction of masses related to La, Fe, and Mn ions caused by the vibration of the oxygen atoms in comparison to the limitation of the harmonic oscillator. The doping of lighter or heavier ions in the La_2_Co_1−z_Fe_z_MnO_6_ compound causes expansion or shrinkage in the size of the whole crystal. Due to the shrinkage, the crystal’s influence on the volume function ultimately gives an upsurge for the variation of the phonon mode frequencies.

### 3.3. Electrical Properties

By adopting the two probe methods, we measured the electrical properties of the double-perovskite La_2_Co_1−z_Fe_z_MnO_6_ (z = 0, 0.2–1.0) ceramics in the temperature range of 303–423 K. From the I–V measurement, we observed a linear response with an increase in the current in all temperature ranges that obeyed Ohm’s law. Furthermore, the current was enhanced with the specific voltage V_o_ = 15V with temperature, displaying the semiconductor behavior of this ceramic. The I–V curves at various temperatures for La_2_Co_1−z_Fe_z_MnO_6_ (z = 0, 0.2–1.0) are shown in Figure 5. It can be observed that the current increases significantly when adding Fe doping, indicating an enhancement of the electrical behavior of the doped samples. This increment behavior may result from creation of polaron generated from the distorted lattice structure by the substitution of Co with Fe [19]. Temperature-dependent resistivity was calculated using the equation:(5)$$\rho =\frac{RA}{L}$$

According to the adiabatic nearest-neighbor hopping over small polarons (Holstein polaron) model (ANHSP), the decline in resistivity with the enhancement in temperature is due to the thermally triggered charges corresponding to the conduction mechanism of hopping [21,22,23], and the activation energy of the samples was calculated based on the formula expressed in Equation (6), with the help of the Arrhenius plot of Lnρ versus 1/*k_β_ T*.
(6)$$E_{a}=k_{\beta }TLn\left(\frac{\rho }{\rho {}^{\circ}}\right)$$
where *β* is the Boltzmann constant, *E_a_* is the activation energy, *ρ* represents the resistivity, and *T* is the absolute temperature. The activation energy of the samples increases with the mole percentage of Fe^+3^ due to the large atomic radii compared to the Co^3+^. Usually, the activation energy (*E_a_*) of the polaronic conduction of holes and electrons is close to 0.2 eV, but in the case of oxides, ionic conductors have *E_a_* greater than 0.90 eV. The calculated activation energies from the Arrhenius plot support the semiconducting nature of all samples and the negative correlation between the resistivity and the iron concentration. The polaronic hopping conduction in this temperature range (303–423 K) is explained by the hopping polaron model [24,25,26,27,28], which explains the bandgap conduction mechanism. Usually, the low-temperature resistivity curve is explained by the electron-phonon scattering or impurity scattering mechanisms by changing the slope of *ρ* (*T*) around a charge carrier’s localization temperature [29]. In the high-temperature regime, small polaron hopping appears to be effective by the linear curve fit, while variable range hopping (VRH) conduction is dominated for the low-temperature regime. The temperature-dependent resistivity (left side) and the Arrhenius law (inset of right side) for La_2_Co_1−z_Fe_z_MnO_6_ (z = 0, 0.2–1.0) is shown in Figure 6. The insets represent the concentration-dependent resistivity and the extracted activation energy from the Arrhenius law. It can be seen that the activation energy increases with Fe doping and reaches a maximum with a 0.5 Fe doping ratio, then suppression may be due to the agglomeration and segregation of Fe nano-additives at grain boundaries.

### 3.4. Dielectric Properties

The dielectric properties of double-perovskite La_2_Co_1−z_Fe_z_MnO_6_ (z = 0, 0.2–1.0) composites were measured by conductance (G) and capacitance (C) with an LCR meter. The real part ($\;\varepsilon_{r}^{\prime }$) and imaginary part ($\varepsilon^{\prime \prime }$) of the dielectric constant, dielectric loss (tanδ), and ac-conductivity (σ*_ac_*) were studied in the frequency range of 10 Hz–1 MHz between 303 and 423 K. The real part of the dielectric constant ($\;\varepsilon_{r}^{\prime }$) displays the capacitive energy storage in the material when the electric field is applied, as shown in Equation (7).
(7)$$\;\varepsilon_{r}^{\prime }=\frac{Cd}{A\varepsilon_{o}}$$
where *ε_o_* represents the permittivity of the free space, *A* is the area, and *d* is the thickness of the pellet. According to the Wagner dielectric response theory, the grain boundaries are nonconducting layers, while the grains are conducting due to the hopping process. The charges move to the poorly conducting layer from the good one with an applied electric field [30,31]. The real part (*Log*$\;\varepsilon_{r}^{\prime }$) of the dielectric constant versus frequency at various temperatures for La_2_Co_1−z_Fe_z_MnO_6_ (z = 0, 0.2–1.0) is shown in Figure 7a–f. The inset of Figure 7a–f represents the temperature-dependent real part of each plot. The values of the real part (*Log*$\;\varepsilon_{r}^{\prime }$) increase with the increase in temperature and Fe concentration due to the relaxation of electric dipoles, but at a higher frequency, they are found to be flat, coinciding with each other close to zero. This reveals that the relaxation procedure ceases at a higher frequency due to the short time scale involved. The sharp fall in real part (*Log*$\;\varepsilon_{r}^{\prime }$) at higher frequency originates from dielectric relaxation. The gradual decline in the real part of the dielectric constant (*Log*$\;\varepsilon_{r}^{\prime }$) in the low-frequency region can be ascribed to space charge polarization due to blocked charge carriers at physical barriers, such as grain boundaries. At low frequencies, grain boundaries or interfacial polarization play a vital role in dominating the dielectric properties of the material. At maximum values of the real part ($\;\varepsilon_{r}^{\prime }$) at low frequency, the charges begin to accumulate at grain boundaries. The net polarization is changed due to the dominance of the space charge polarization [32]. The increase in the dielectric parameters (*Log*$\;\varepsilon_{r}^{\prime }$) with increasing z-contents may arise due to the surface volume fraction of Fe^3+^ ions compared to the cobalt Co^3+^ ions. The radius of the Fe^3+^ is (0.645 Å) in a high-spin state is relatively higher than the Co^3+^ ions (0.61 Å) in a low-spin state [14,33,34], which enhances the barrier for charge mobility at the grain boundaries and plays an important role in the space charge polarization. The contribution of net dipoles becomes slow at higher frequencies, and only electronic polarization remains active due to the very small net polarization in comparison to low-frequency polarization. Similarly, at 10 Hz and 423 K, the peak values of the real part (*Log*$\;\varepsilon_{r}^{\prime }$) are 10.60, 10.61 11.47, 10.99, 10.91, and 10.01 with concentrations (z = 0, 0.2–1.0). It is found that the magnitude of the dielectric constant (*Log*$\;\varepsilon_{r}^{\prime }$) increases with temperature and Fe concentrations (z = 0, 0.2–1.0). Due to increase in temperature, space charges are released, which participate in polarization. Grain and grain boundaries are also increased in size due to an increase in temperature.
(8)$$\varepsilon^{\prime \prime }=\frac{Gd}{A\omega }$$

In the same temperature and frequency range, the imaginary part (*Log ε^″^*) is also calculated by using Equation (8). By applying an AC field, the imaginary part of the dielectric displays the energy dissipated within the material [35]. The variation in the imaginary part versus frequency at selected temperatures from 303 to 423 K with different concentrations of z = 0, 0.2–1.0 composites is shown in Figure 8a–f. The estimated values of the imaginary part (*Log ε^″^*) versus temperature are represented in the inset of each graph with z = 0, 0.2–1.0. The imaginary part displays higher values at low frequency and is found to be constant at high frequency. According to the Maxwell–Wagner model and Koop’s theory, the dielectric medium is assumed to be made of well-conducting grains separated by grain boundaries.

As the imaginary part (*Log ε^″^*) depends on the conductance, none of the electrons in the dielectric ceramic are bound. These free electrons play a vital role in conductance. This current loss may be due to the migration of free electrons or any other energy consumption, whereas only ion vibration losses contribute at high frequencies [9,36]. The imaginary part of the dielectric constant (*Log ε^″^*) shows no response at a higher frequency due to a short time period. Most carriers cannot follow the external electric field, and the material response becomes very small. Grain boundaries are more effective at low frequency, due to which charges hop, resulting in a higher dielectric constant, and the carriers can follow the frequency of externally applied ac field.

The dependence of the dielectric loss or tanδ on the frequency at different temperatures for the double perovskites is shown in Figure 9(a–f). Tangent loss is generally referred to as a dissipation factor and calculated from Equation (9), and it is significant to the mechanisms of dielectric relaxation and ac conduction.
(9)$$\tan\delta =\frac{\varepsilon^{\prime \prime }}{\varepsilon_{r}^{\prime }}$$

The dissipation of energy (tanδ) shows maximum values with the temperature increase at low frequencies. The tanδ has a higher value compared to the energy stored at low frequencies. The tanδ decreased with increasing frequency due to the attributed charges to the field [37]. The variations in tanδ were observed with frequency for different concentrations of iron. The dielectric loss increased with increasing the concentration, and temperature variation is also shown in each inset of Figure 10a–f. The energy losses may be attributed to the collision, vibration, and other charges’ phenomena. Moreover, the loss is related to the period difference between relaxation and the applied field. Low relaxation time results from the instantaneous polarization process while large relaxation time is associated with the delayed polarization process. When the applied field has the same period of relaxation, the loss contribution is at its maximum. Losses turn smaller when the relaxation time greatly differs between the period of the applied field. The loss becomes small if the relaxation time is relatively larger than the applied field frequency. In this situation, the polarization mechanism is much slower than the reversal of the field, which cannot make ions follow the applied field. Similarly, if the relaxation time is much smaller than the applied field frequency, the polarization process can easily follow the applied field with no lag, creating a small loss.

The σ*_ac_* versus Log f of La_2_Co_1−z_ Fe_z_MnO_6_ (z = 0, 0.2–1.0) for the temperature range 303–423 K is plotted in Figure 11a–f. The A.C σ*_ac_* is estimated with Equation (10)
(10)$$\sigma_{\mathrm{ac}}=\omega \varepsilon_{0}{\varepsilon^{\prime }}_{r}\tan\delta$$

The σ*_ac_* values remained almost constant for all samples at low frequency and showed an abrupt increase at high frequency. At low frequency, the hopping of charges across the grain boundaries is minimal and increases with frequency. The temperature-dependent AC conductivity is shown in the inset of each plot in Figure 11a–f. The values of σ*_ac_* also increased with the temperature because charge carriers can more easily hop the barrier at high temperatures compared to low temperatures [38]. The other reason is that large ionic radii Fe^+3^ (0.645 Å) in a high-spin state with relatively higher surface effects than the Co^3+^ ion (0.61 Å) in a low-spin state pile up the free carriers at the grain boundaries, where these carriers have enough energy to jump over the barriers at high frequencies.

## 4. Conclusions

A series of double-perovskite La_2_Co_1−z_Fe_z_MnO_6_ (z = 0., 0.2–1.0) composites were synthesized successfully by the sol–gel technique. The structural properties were investigated by XRD and confirmed that the synthesized composites possess a monoclinic phase with a P2_1_/n symmetry at room temperature. The obtained IR spectra were fitted well by the Lorentz oscillator model to calculate the damping factor, optical frequency, and oscillator strength and compared with the theory, which gave better agreement. Moreover, I–V measured the curve at room temperature following Ohm’s law. The calculated activation energies from the Arrhenius plot support the semiconducting nature of all samples. The resistivity decreases with increased concentration. Furthermore, the frequency and temperature-dependent dielectric constant ($\;\varepsilon_{r}^{\prime },\;\varepsilon^{\prime \prime }$) and dielectric loss (tanδ) were enhanced at low frequencies while the ac-conductivity (σ_ac_) displayed the maximum value at high frequencies. The increase in the dielectric parameters with increasing z-contents may arise due to the surface volume fraction of Fe^3+^ ions compared to the cobalt Co^3+^ ions. The radius of the Fe^3+^ is (0.645 Å) is relatively higher than the Co^3+^ ions (0.61 Å), which can be significantly influenced by the grains and grain boundaries, enhancing the barrier for charge mobility at the grain boundaries that play an important role in the space charge polarization.

## Figures and Tables

**Figure 1 materials-15-06249-f001:**
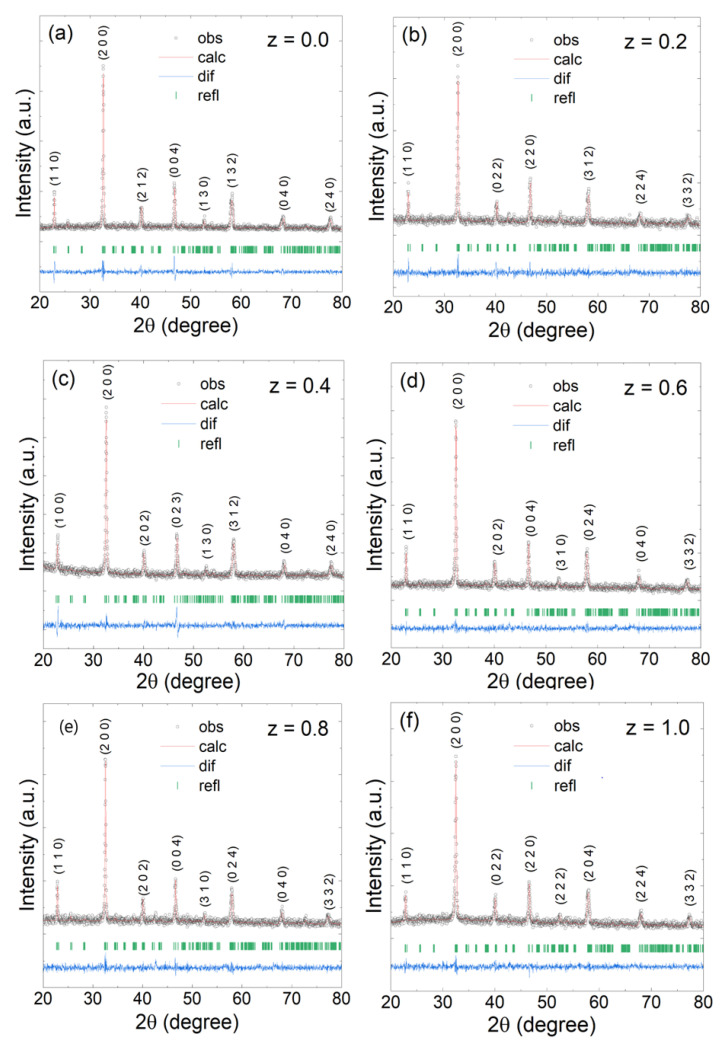
The X-ray diffraction scans of (**a**) La_2_CoMnO_6_ (**b**) La_2_Co_0.8_Fe_0.2_MnO_6_ (**c**) La_2_Co_0.6_Fe_0.4_MnO_6_ (**d**) La_2_Co_0.4_Fe_0.6_MnO_6_ (**e**) La_2_Co_0.2_Fe_0.8_MnO_6_ (**f**) La_2_FeMnO_6_.

**Figure 2 materials-15-06249-f002:**
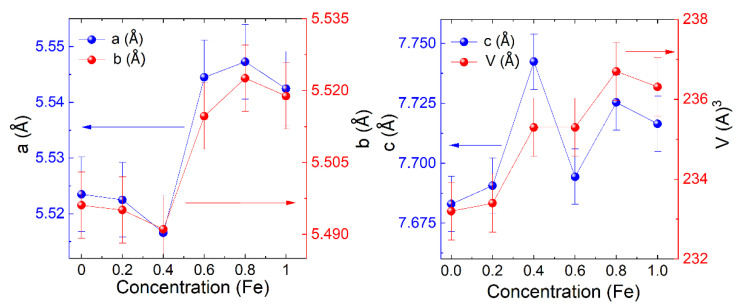
The plot of lattice parameters versus contents.

**Figure 3 materials-15-06249-f003:**
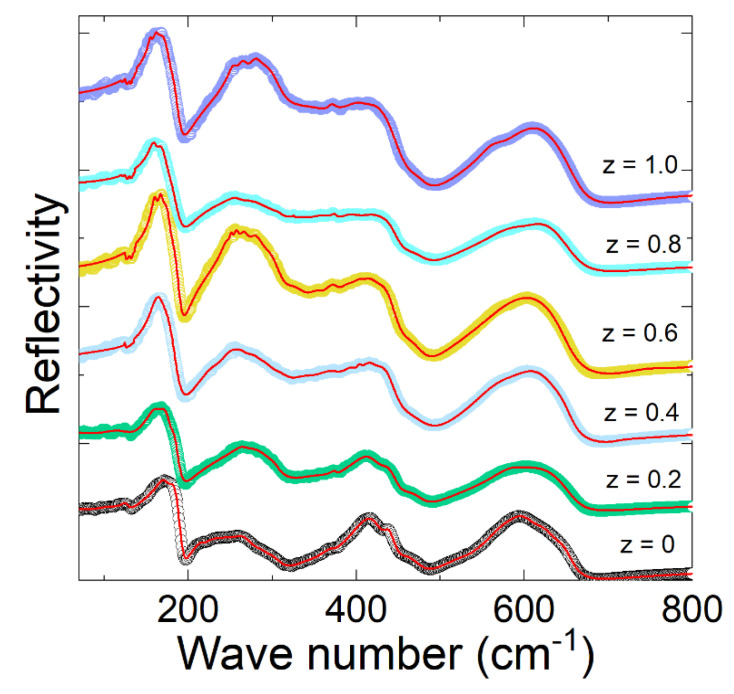
IR spectra of La2Co1−zFezMnO6 (z = 0, 0.2–1.0) at room temperature: red line displays experimental values and black circle represent the refined fitted data.

**Figure 4 materials-15-06249-f004:**
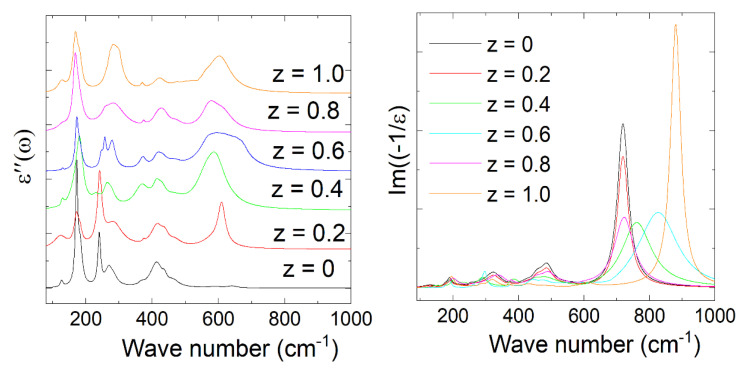
The imaginary parts of (**left** side) the dielectric function of the La_2_Co_1__−__z_Fe_z_MnO_6_ (z = 0, 0.2–1.0) ceramics at room temperature in the far-infrared regions and (**right** side) the reciprocal dielectric function.

**Figure 5 materials-15-06249-f005:**
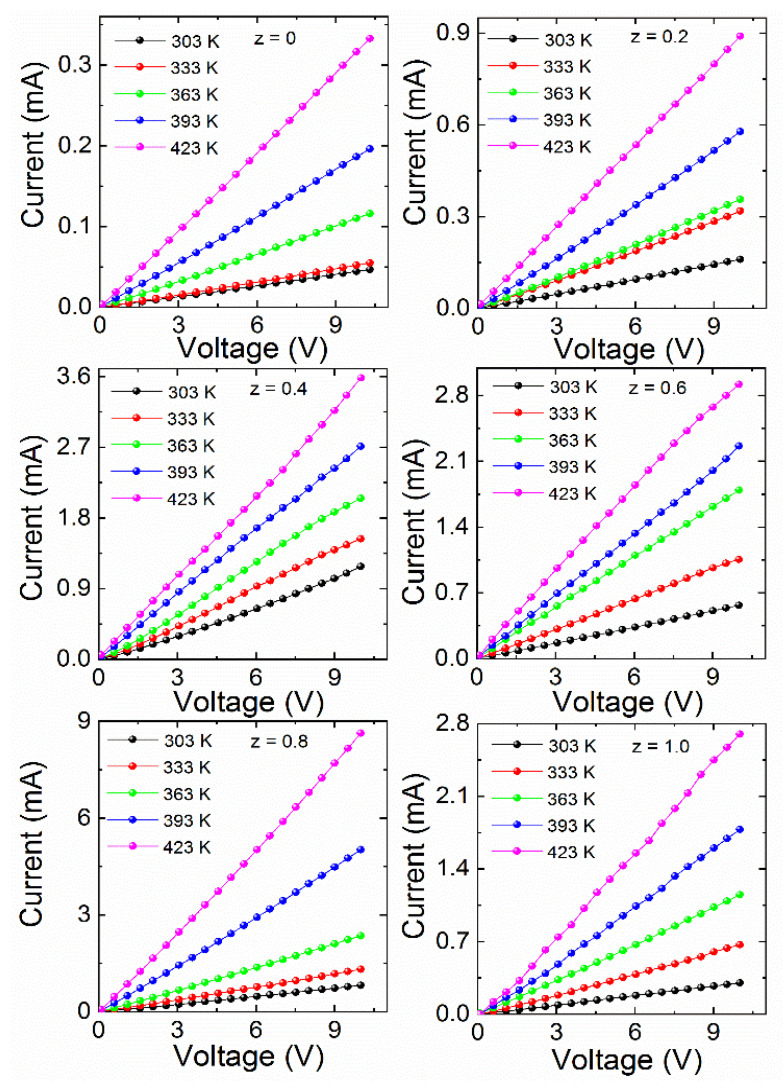
I–V curves at various temperatures for La_2_Co_1−z_Fe_z_MnO_6_ (z = 0, 0.2–1.0).

**Figure 6 materials-15-06249-f006:**
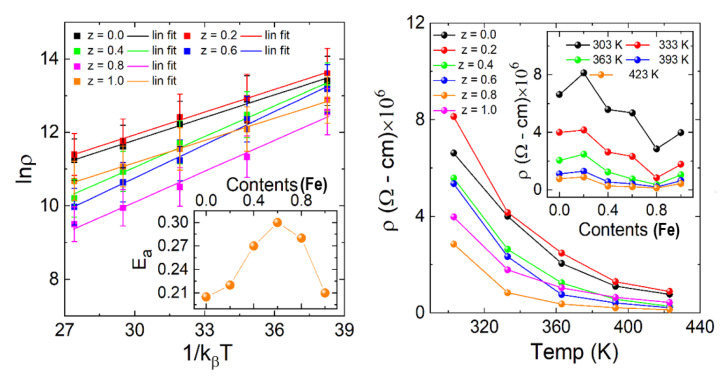
Temperature-dependent resistivity (**left** side) and Arrhenius law (**right** side) for La_2_Co_1−z_Fe_z_MnO_6_ (z = 0, 0.2–1.0). Insets represent the concentration-dependent resistivity and calculated activation energy.

**Figure 7 materials-15-06249-f007:**
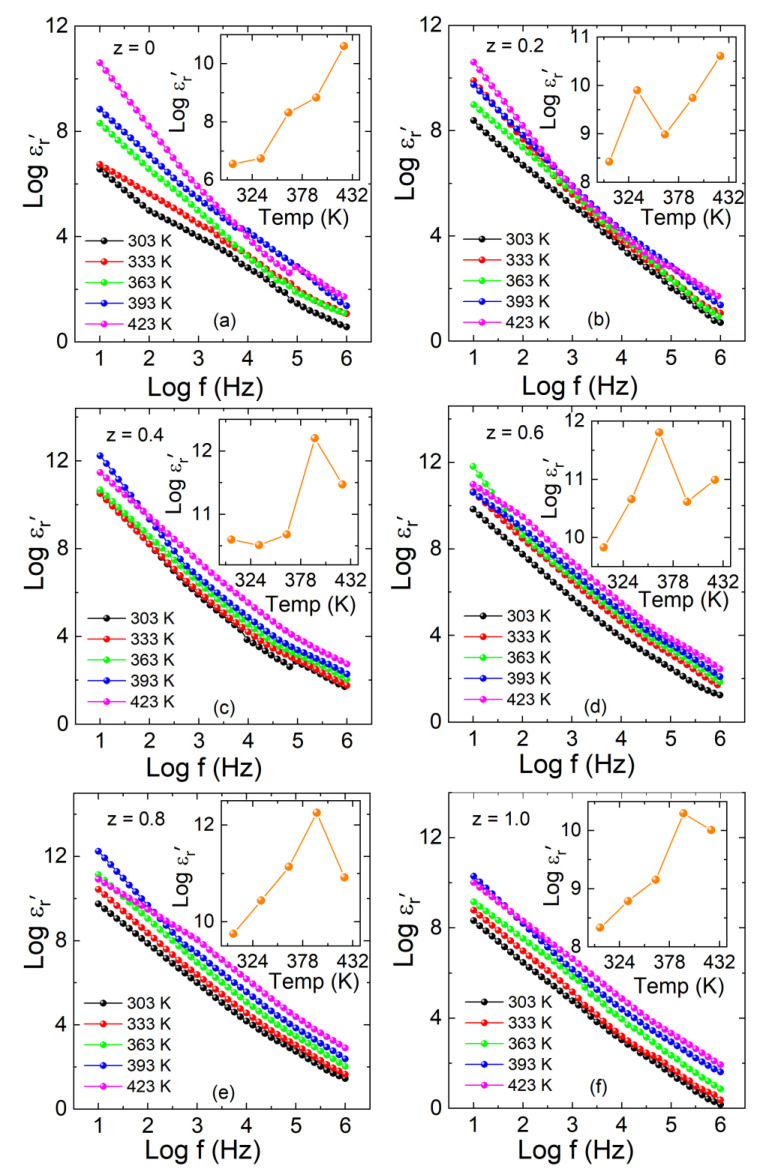
(**a**–**f**) The real part of the dielectric constant (*Log*$\;\varepsilon_{r}^{\prime }$) versus Log f graph for La_2_Co_1−z_Fe_z_MnO_6_. Inset of (**a**–**f**) represents the temperature-dependent real part of each plot.

**Figure 8 materials-15-06249-f008:**
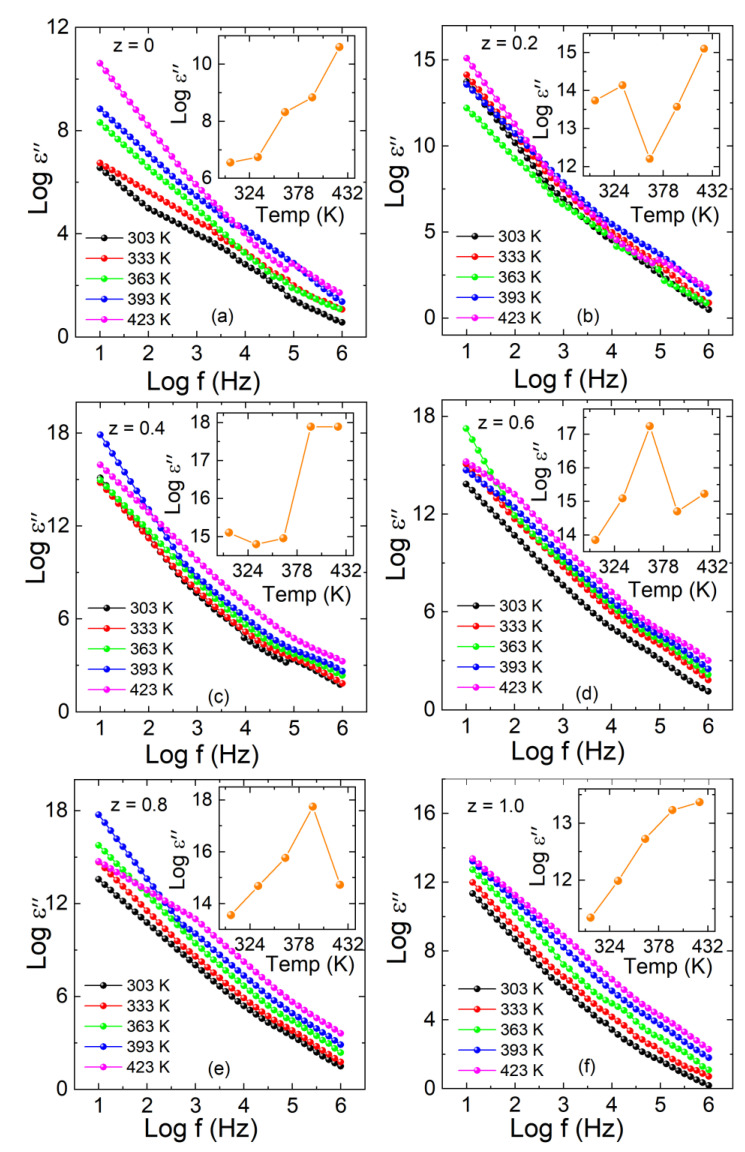
(**a**–**f**) Imaginary part of dielectric (*Log ε^″^*) versus Log f graph for La_2_Co_1−z_Fe_z_MnO_6_ (z = 0, 0.2–1.0). Inset of (**a**–**f**) represents the temperature-dependent imaginary part of each plot.

**Figure 9 materials-15-06249-f009:**
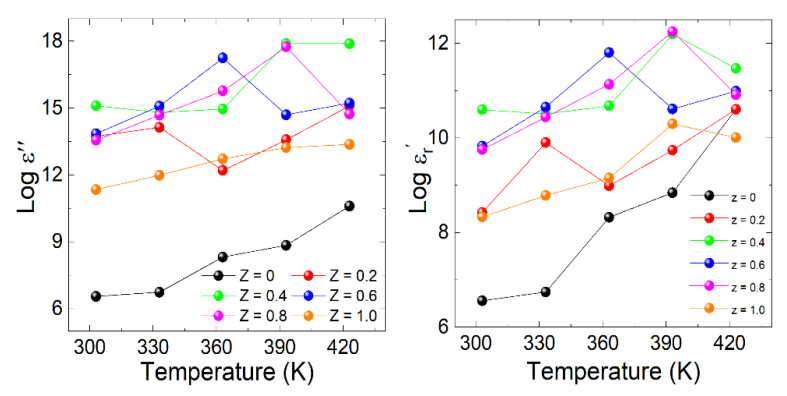
The variation of the real part of the dielectric constant (*Log*$\;\varepsilon_{r}^{\prime }$) (**left** side) and Imaginary part of dielectric (Log ε^″^) (**right** side) for La_2_Co_1−z_Fe_z_MnO_6_ (z = 0, 0.2–1.0).

**Figure 10 materials-15-06249-f010:**
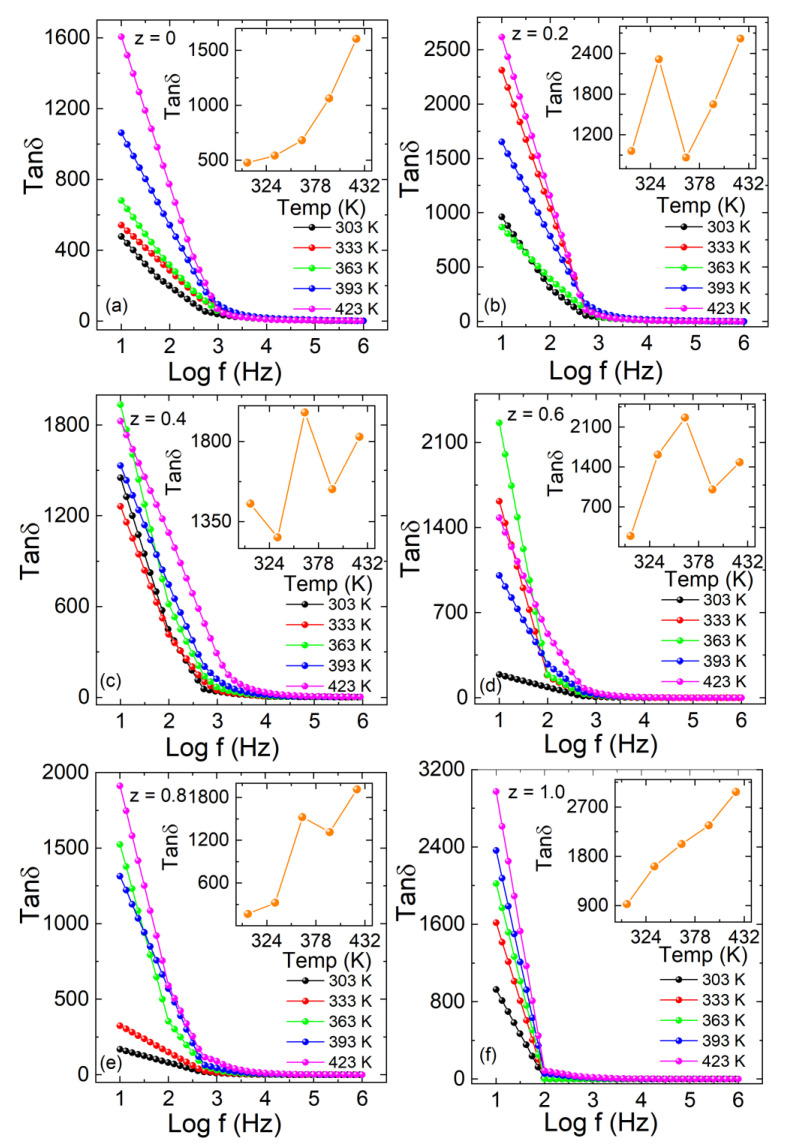
(**a**–**f**) Dielectric loss (tanδ) versus Log f graph for La_2_Co_1−z_Fe_z_MnO_6_ (z = 0, 0.2–1.0). Inset of (**a**–**f**) represents the temperature-dependent dielectric loss part of each plot.

**Figure 11 materials-15-06249-f011:**
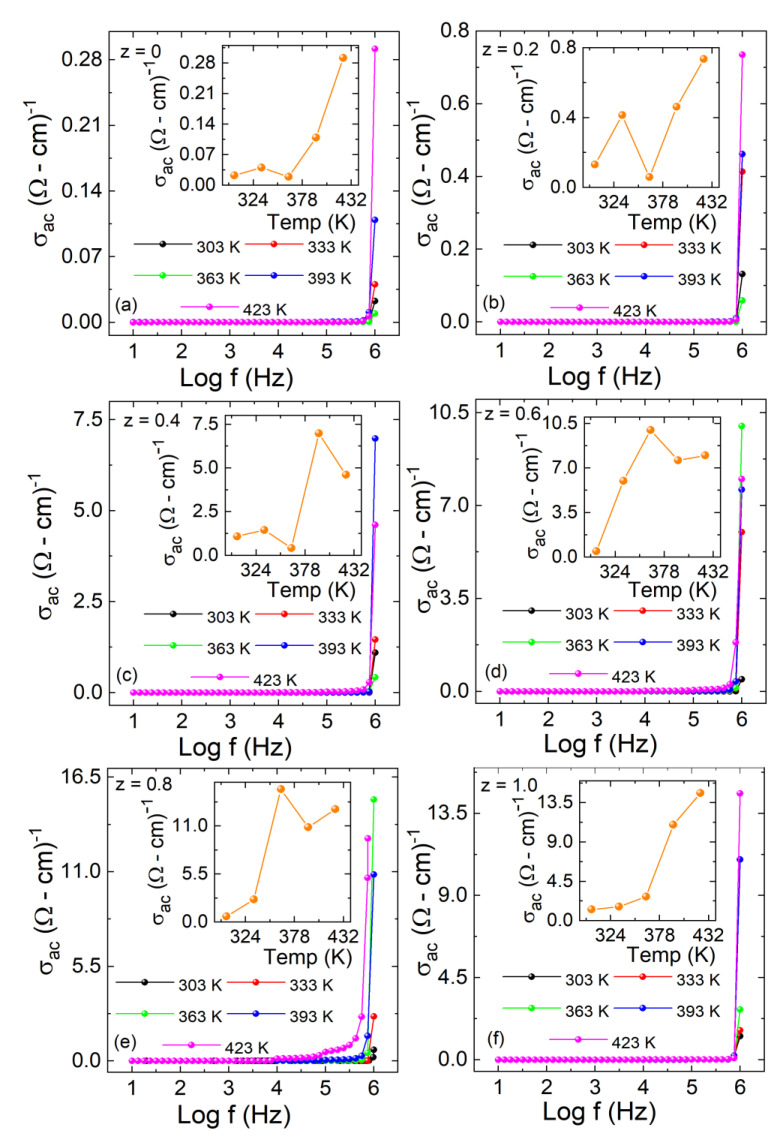
(**a**–**f**) Ac conductivity (σ*_ac_*) versus Log f graph for La_2_Co_1−z_Fe_z_MnO_6_ (z = 0, 0.2–1.0). Inset of (**a**–**f**) represents the temperature-dependent Ac conductivity part of each plot.

**Table 1 materials-15-06249-t001:** Refined lattice parameters of La_2_Co_1−z_Fe_z_MnO_6_ (z = 0, 0.2–1.0).

Material	Z = 0	Z = 0.2	Z = 0.4	Z = 0.6	Z = 0.8	Z = 1.0
S. G	P2_1_/n	P2_1_/n	P2_1_/n	P2_1_/n	P2_1_/n	P2_1_/n
a (Å)	5.5235	5.5225	5.5166	5.5445	5.5473	5.5424
b (Å)	5.4961	5.4951	5.4911	5.5147	5.5226	5.5189
c (Å)	7.6830	7.6907	7.7424	7.6944	7.7254	7.7165
V (A)^3^	233.2	233.4	235.3	235.3	236.7	236.3
β	90.30	89.78	90.62	89.70	90.40	90.31
R_p_	6.34	10.40	7.28	8.77	10.83	8.42
R_wp_	8.03	13.27	9.30	11.28	13.74	10.67
GOF	0.91	0.86	0.92	0.91	0.86	0.87
Grain size (nm)	83	84	81	82	81	80

**Table 2 materials-15-06249-t002:** Fitted parameters of optical transverse phonons of (ω_TO_) at room temperature.

ω_TO_ (cm^−1^)	Z = 0	Z = 0.2	Z = 0.4	Z = 0.6	Z = 0.8	Z = 1.0
ω_TO1_	127.10	124.83	130.83	129.10	128.75	127.40
ω_TO2_	172.30	172.06	173.55	172.90	168.10	168.90
ω_TO3_	183.10	182.94	181.82	182.56	178.61	182.24
ω_TO4_	240.44	241.90	231.07	246.40	-	-
ω_TO5_	268.82	264.81	263.92	257.54	257.82	279.35
ω_TO6_	283.72	285.78	278.33	279.33	286.44	293.97
ω_TO7_	366.8	374.52	369.39	371.78	374.31	370.05
ω_TO8_	413.41	414.79	412.96	414.76	419.40	422.03
ω_TO9_	437.57	438.48	430.12	430.9	435.95	-
ω_TO10_	465.09	469.81	467.49	466.6	472.02	472.88
ω_TO11_	-	-	563.29	568.21	569.76	562.23
ω_TO11_	589.91	583.29	593.51	571.75	583.47	594.84
ω_TO12_	641.22	609.61	642.18	637	615.79	612.06

**Table 3 materials-15-06249-t003:** Refined experimental parameters of strength S_TO_ of transverse optical phonon modes for La_2_Co_1−z_Fe_z_MnO_6_ (z = 0, 0.2–1.0) ceramic.

Z	Z = 0	Z = 0.2	Z = 0.4	Z = 0.6	Z = 0.8	Z = 1.0
S_1_	0.061	0.540	1.250	0.016	0.127	0.257
S_2_	0.680	0.373	0.585	0.741	0.622	1.771
S_3_	0.426	0.105	0.327	0.455	0.649	0.258
S_4_	0.273	0.173	0.134	0.139	0.135	-
S_5_	0.197	0.345	0.222	0.241	0.038	0.517
S_6_	0.115	0.712	0.125	0.626	1.012	0.0.58
S_7_	0.015	0.005	0.383	0.185	0.011	0.017
S_8_	0.339	0.283	0.133	0.169	0.162	0.147
S_9_	0.025	0.065	0.111	0.184	0.107	-
S_10_	0.053	0.067	0.137	0.122	0.075	0.025
S^/^_11_	-	-	0.377	0.142	0.158	0.069
S_11_	0.011	0.141	0.123	0.650	0.151	0.540
S_12_	0.013	0.233	0.015	1.760	0.223	0.740

**Table 4 materials-15-06249-t004:** Calculated parameters of damping factor γ_TO_ of transverse optical phonon modes for the La_2_Co_1−z_Fe_z_MnO_6_ (z = 0, 0.2–1.0) at room temperature.

Z	z = 0	z = 0.2	z = 0.4	z = 0.6	z = 0.8	z = 1.0
γ_1_	9	45	41	34	30	26
γ_2_	08	16	23	10	14	18
γ_3_	15	12	20	16	28	16
γ_4_	10	13	30	11	16	-
γ_5_	25	17	25	11	17	21
γ_6_	30	65	26	23	82	35
γ_7_	17	7	53	25	09	11
γ_8_	44	44	28	29	41	45
γ_9_	15	26	32	36	36	
γ_10_	38	45	68	50	49	34
γ^/^_11_	-	-	73	44	51	34
γ_11_	46	78	71	88	61	83
γ_12_	35	27	57	61	70	76

## Data Availability

The data presented in this study are available on request from the corresponding author. The data are not publicly available to make sure of their proper usage.

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
