# Peer review of "Structural, Optical, Charge-Transport, and Dielectric Properties of Double-Perovskite La2Co1−zFezMnO6 (z = 0, 0.2–1.0)"

_materials, 2022, doi:10.3390/ma15186249_

Round 1

Reviewer 1 Report

The manuscript is very interesting. Some revision should be performed though:

* References are quite old. The "newest" is from 2018. Re-check if some new information is available.

* Experiments section is incomplete. More details about the synthesis and performed experiments should be provided.

* Please, unify the general format of the text. Check the english language. Some sentences mix the verb tense.

* Check the attached PDF file.

The manuscript was read several times with dedication and in detail. All the observations, comments and suggestions are made with all do respect to the authors and their work.

Author Response

Kindly see the attachment

Reviewer 2 Report

The authors of this study synthesized a ferromagnetic double perovskite series and calculated some optoelectronics properties to validate their synthesis. Although the material itself gained some interest in last decade, but I think I can not recommend this paper in its present form. My comments are as follows:

1.     In Introduction part, the authors must add why did they choose to explore properties such as phase, optical, charge transport, and dielectric etc.

2.     The experiment section needs more explanation. It is too much general. A detail discussion about all experimental setups must be included. What was the resolutions of the FTIR spectrometer, and spectral features?

3.     The authors must add some analytical discussion about the I-V curves (Fig. 4). What is the origin of difference between the six plots from a to f?

4.     Similar discussions on Fig. 6 and 7 will increase the quality and novelty of the ms.

5.     Tangent loss is generally referred to as a dissipation factor and calculated from Equation (8), and it is significant to the mechanisms of dielectric relaxation and ac conduction.” Elaborate this statement with appropriate background references.

6.     What are the ionic and electronic contributions to the relative dielectric permittivity? I cannot see them in the ms.

7.     I cannot see any first principles calculations, and any results obtained from those calculations. This means that the experimental results reported in this study were not validated by computational methods. So, it is difficult to accept whether the data provided in this ms are reliable!!

8.     Table captions read very wired. What did the authors mean by “Obtained paramaters …
“? How did they obtain them?

9.     Each lattice property given in Table 1 should be described properly. Otherwise, it is difficult to follow.

10.  Only TO modes are shown, but LO modes are missing. Is there any role that was played by anharmonicity? Is there any coupling between the LO and TO modes? Where are the phonon dispersion curves? Was inelastic scattering experiment performed? Are the phonon modes reported correspond to the zone-center?

Round 2

Reviewer 2 Report

The authors of this study revised a their ms to based on reviewer's  comments to some extent. However I still have some more concerns and I can not recommend this ms now. 

1. Their answer to comment 10 is unclear. The authors must clearly state this point in ms. Phonon dispersion curve is very much important to understand the stability of the material.  

2. Did the authors checked any mechanical properties of the material? Can they shed any light on mechanical stability of the material?

On a small note please prepare the pdf without the comment panel from nect time. 

Author Response

Previous Question

  1. Only TO modes are shown, but LO modes are missing. Is there any role that was played by anharmonicity? Is there any coupling between the LO and TO modes? Where are the phonon dispersion curves? Was an inelastic scattering experiment performed? Are the phonon modes reported correspond to the zone center?

New Revision

The authors of this study revised a their ms to based on reviewer's comments to some extent. However I still have some more concerns and I can not recommend this ms now. 

  1. Their answer to comment 10 is unclear. The authors must clearly state this point in ms. Phonon dispersion curve is very much important to understand the stability of the material. 

[Reply] Now, the transverse phonon modes and longitudinal modes (LO) are extracted from the experimental data and added to the manuscript, which can be seen in figure1 & 2.

Figure 1. IR spectra of La2Co1-zFezMnO6 (z = 0.0, 0.2-1.0) at room temperature red line displays experimental values and black circle represent the refine fitted data.

  • (b)

Figure 2. (a)The imaginary parts of the dielectric functions of the La2Co1-zFezMnO6 (z = 0.0, 0.2-1.0)  ceramic in the far infrared region; (b) imaginary part of the reciprocal dielectric function.

Yes, as discussed in the literature, changes in the cationic structural modification mainly modify the phonon anharmonicity, a disorder of the samples, the wider their bands are in the spectrum since the phonon lifetime along the lattice is small. The dielectric losses are large, the spectra of the ordered and partially disordered LCMO are highly similar [1], and the modes can be assigned based on the work of Linka et al. [2].

The transversal and longitudinal phonon branches, dielectric strengths, and damping of the strongest dielectric modes support the significant contribution of the phonon modes to the structural phase transition (SPT) and reveal an important lattice anharmonicity, especially for the low-frequency modes. Also, these investigations showed that structural ordering does not inhibit the SPT and provided valuable information regarding the polar phonons and their effects on the intrinsic dielectric constant in double perovskites and related compounds.

No results were reported from the inelastic scattering experiment. The complex dielectric function e (w) is expressed in terms of IR-active phonons [3-5]. The spectra are dominated by four bands that can be resolved (by fitting) into 13 polar modes. At room temperature, LCMO crystallizes into a monoclinic structure that belongs to the P21/n (ITA number # 14 or C52h) space group. For this structure, the number and the activities of the first-order vibrational modes can be predicted using group theory tools, such as the nuclear-site method of Rousseau et al. [6]. Once the occupied Wyckoff positions are known, the phonon modes at the Γ point of the first Brillouin zone centre in this phase can be distributed according to the 2/m(C2h) factor group, as presented in Table 1.

Table 1

Normal vibrational modes of the  LCMO at the Brillouin zone center (Г-point), within the P2/n,(14 or ) monoclinic space group with Z = 2.

Atom

Ox. State

Wyckoff position

Occup. fraction

Site symmetry

Irreducible representation

La

3+

4e

1

C1

3Ag Å 3Au Å 3Bg Å 3Bu

Co

2+

2c

1

Ci

3Au Å3Bg

Mn

4+

2d

1

Ci

3Au Å3Bg

O(1)

2-

4e

1

C1

3Ag Å 3Au Å 3Bg Å 3Bu

O(2)

2-

4e

1

C1

3Ag Å 3Au Å 3Bg Å 3Bu

O(3)

2-

4e

1

C1

3Ag Å 3Au Å 3Bg Å 3Bu

Γ IR =17Au Å 16Bu

Γ Raman=12Ag Å 12Bg

Γacoustic=Au Å 2Bu

 Therefore, according to the group’s theoretical predictions, 33 IR active modes (17Au ⊕ 16Bu) are expected for monoclinic LCMO. The number of observed polar phonons is lower than that predicted by the group theory, as it is typically observed in monoclinic ceramic samples [6-8].

  1. Did the authors checked any mechanical properties of the material? Can they shed any light on mechanical stability of the material?

No, but structural and magnetic properties, resistivity, and magnetoresistance effects of double perovskite La2Co1-xFexMnO6 with the almost same doping ratio already reported in the literature. (http://dx.doi.org/10.1016/j.jallcom.2015.11.097)

On a small note please prepare the pdf without the comment panel from nect time. 

References

[1] R.X. Silva, A.S. Menezes, R.M. Almeida, R.L. Moreira, R. Paniago, X. Marti,H. Reichlova, M. Maryško, M.V.S. Rezende, C.W.A. Paschoal, J. Alloys Compd. 661 (2015) 541–552.

[2] J. Hlinka, J. Petzelt, S. Kamba, D. Noujni, T. Ostapchuk, Phase Transit. A Multinatl.
J. 79 (2006) 41–78.

[3] A. Dias, G. Subodh, M.T. Sebastian, R.L. Moreira, J. Raman Spectrosc. 41 (2009)
702–706.
[4] R.X. Silva, R.L. Moreira, R.M. Almeida, R. Paniago, C.W.A. Paschoal, J. Appl. Phys.
117 (2015) 214105.

[5] F. Gervais, B. Piriou, Phys. Rev. B Solid State 10 (1974) 1642–1654

[6] D.L. Rousseau, R.P. Bauman, S.P.S. Porto, J. Raman Spectrosc. 10 (1981) 253–290.

[7] D. Augusto, B. Barbosa, Spectrochimica Acta A 185 (2017) 125–129.

[8] J. Hlinka, J. Petzelt, S. Kamba, D. Noujni, T. Ostapchuk, Phase Transit. A Multinatl.
J. 79 (2006) 41–78.
